# Generalizing to Unseen Domains for Regression

## Abstract

In the context of classification, *domain generalization* (DG) aims to predict the labels of unseen target-domain data using only labeled source-domain data, where the source and target domains usually share *the same label set*. However, in the context of regression, DG is not well studied in the literature, and the main reason is that the ranges of response variables in two domains are often *different*, even disjoint under some extreme conditions. In this paper, we systematically investigate domain generalization in the regression setting and propose a weighted meta-learning strategy to obtain optimal initialization across domains to tackle the challenge. Unlike classification, the labels (responding values) in regression naturally have ordinal relatedness. The relatedness brings a core challenge in meta-learning for regression: the hard meta-tasks with less ordinal relatedness are under-sampled from training domains. To further address the hard meta-tasks, we adopt the feature discrepancy to calculate the discrepancy between any two domains and take the discrepancy as the importance of meta-tasks in the meta-learning framework. Extensive regression experiments on the standard benchmark DomainBed demonstrate the superiority of the proposed method.

## 1  Introduction

*Domain generalization* (DG) receives increasing attention due to its challenging setting: learning models on source domains and inferring on unseen but related target domains [1, 2]. However, most existing approaches focus on semantically invariant representations for classification, limiting their practical applications to regression tasks. For example, real-world applications often involve predicting the recovery/survival time of patients in clinic or estimating the ages/skeleton joints/gaze direction of humans [3, 4, 5]. These tasks can be grouped into cross-domain regression problems.

In cross-domain regression, the label's marginal distribution shift can differ significantly compared to DG for classification. In DG classification, the shift typically represents variations in class probability densities across domains [6]. In regression, the shift can take on a specific form, e.g., when the responding (regression) interval of the source domain is $[0, 0.7]$, the shifted responding interval of the target domain can be $[0.5, 1]$. This type of shift often occurs in regression settings such as predicting unseen ages, depths and rentals. In some cases, these regression intervals even have no overlap. We refer to this particular regression scenario as *domain generalization in regression* (DGR). Fig. 1 illustrates the differences between imbalanced domain regression and the DGR. Unlike imbalanced regression [7], DGR focuses on exploration or interpolation for regression.

**Comparisons to traditional DG.** From the perspective of domain generalization, DGR can be viewed as a special generalization case where the target labels are continuous. However, most domain generalization methods are suboptimal for addressing the DGR problem due to the ordinal relatedness of regression labels. For example, feature alignment [8] might be unnecessary and even harmful in our DGR setting. Assuming that a closer feature discrepancy implies closer predictions, feature alignment methods may cause the model to exclusively map all predictions into one source interval, which

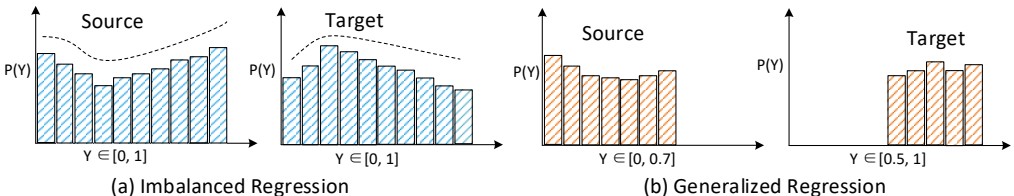

Figure 1: The label distributions of two different regression settings. (a) In the imbalanced domain regression, the response values $Y \in [0, 1]$ exhibit varying probability densities across domains. (b) The DGR problem focuses on predicting unseen response values in the target domain. The response values might encompass both overlapping (just like source interval $[0, 0.7]$ and target interval $[0.5, 1]$) and non-overlapping intervals.

does not reduce total generalization risks. In addition to feature alignment, feature disentanglement usually disentangles semantically related discriminant representation for classification [9], while overlooking the ordinal relatedness of the target domain. Furthermore, semantic-related discriminant representation might be unnecessary for regression tasks like age estimation. Robust optimization methods [10] can perform moderately distributional exploration, but also lack the ability to tackle ordinal relatedness in regression.

**Comparisons to open-set DG [1, 11].** Open-set DG primarily focuses on classification applications and the ability to detect unknown classes. If open-set DG methods are used to address our problem, they can only identify these samples whose response intervals differ from that of the source domain but *cannot* obtain their response values.

To effectively capture ordinal relations and facilitate modest extrapolation in the DGR problem, we propose a robust optimization algorithm via meta-learning. Meta-learning algorithms, e.g., *model agnostic meta-learning* (MAML, [12]) have been extensively utilized in traditional domain generalization [13, 14, 15]. In each meta-task, these methods usually sample a support and a query classification task from two distinct domains and optimize the meta-model by a bi-level paradigm. However, this paradigm alone falls short for addressing the complexities of DGR. The task sampling strategy employed in these methods typically follows an implicit assumption, assuming that all training meta-tasks have equal importance [16, 17]. We argue that this implicit assumption no longer holds in our regression setting.

In contrast to classification, regression tasks exhibit ordinal relations between each pair of labels [18]. When considering the label discrepancy between the support and query domains, it is observed that *meta-tasks with a larger regression margin are sampled less frequently compared to those with a smaller margin.* Additionally, meta-tasks with a larger regression margin tend to be more challenging to optimize within the meta-learning framework. *These key factors bring a sampling bias that harder meta-tasks are less sampled from training data.* Consequently, the sampling bias makes harder meta-tasks underrepresented in the training data, i.e., the meta-model tends to choose the easier meta-tasks, limiting the exploration and interpolation capabilities of the model. To mitigate this sampling bias, we propose a simple yet effective strategy: assigning higher weights to harder meta-tasks. These weights are computed based on the feature discrepancy between the query and support examples of each meta-task.

In conclusion, we have developed a DGR benchmark that encompasses both overlapping and non-overlapping labels between the source and target domains. We conduct experiments on three regression tasks, including causality exploration with a toy logic dataset, predicting unseen ages according to face images, and forecasting rental prices across different regions. Our proposed method, named *margin-aware meta regression* (MAMR), makes the following main contributions:

- We investigate generalized regression from the perspective of domain generalization, a previously understudied area with significant practical implications.

- To enhance exploration and interpolation capabilities, we introduce a margin-aware meta-learning framework that mitigates sampling bias and encourages the model to recognize long-range ordinal relations.

- Although our solution achieves considerable improvements regarding baselines, our empirical analyses demonstrate that generalizing to unseen responses is still challenging.

## 2    Related Work

In this section, two related research areas are briefly introduced. One is domain adaptation for ordinal regression and classification, and the other one is generalization for regression.

### 2.1    Domain Adaptation for Ordinal Regression and Classification

Domain adaptation aims to migrate the knowledge from a source domain to a target domain, where there may exist a distribution shift between them. Typical domain adaptation methods try to get confident decision boundaries for classification tasks based on clustering assumption [19]. However, when it comes to cross-domain regression (also known as ordinal classification [18]), these assumptions are not satisfied, posing challenges for existing domain adaptation methods. Some pioneer works like [20] try to provide regression discrepancy in reproducing kernel Hilbert space. Most recent works address cross-domain regression in specific application scenarios, such as estimating object boxes in cross-domain/few-shot object detection [21, 17], regressing human skeleton key-points in cross-domain gesture estimation [4] and calculating the gaze direction in cross-domain gaze tracing [22]. Furthermore, [3] proposes a general cross-domain regression method via subspace alignment, which reduces domain gap by minimizing *representation subspace distance* (RSD) with the principal angles of representation matrices. [23] proposes an adversarial dual regressor to achieve a direct alignment between two domains.

However, nearly all cross-domain regression methods inherently assume there only exists covariate shift in input examples, i.e., $p(x_s) \neq p(x_t)$, where $p(\cdot)$ is the probability density function and $x_s, x_t$ denote the source and target examples. This assumption implies that these methods may not be capable of handling label shift across domains. The label shift in cross-domain regression can arise as interval shift of responding values, e.g., the source interval $y_s \in [0.3, 0.5]$ while the target interval $y_t \in [0.6, 0.7]$. The responding values in the real world can be gasoline consumption data and vary significantly across developed and developing countries [24]. [25] also considers the interval shift problem and tries to learn a ranking on the target domain, followed by mapping the ranking to responding values. This method assumes the availability of the responding interval on the target domain at the adaptation stage, which might be contradictory to the setting of unavailable labels. In contrast, we assume all target domain data are not available at the training stage, which is more practical and challenging in real-world scenarios.

### 2.2    Generalization/Causality for Regression

Domain generalization introduces a more challenging setting where the model can only access the labeled source data at the training stage [1, 2, 26, 27, 28, 29, 30, 31]. A thorough discussion of domain generalization might exceed the scope of our paper. We focus on potential methods that can be applied to regression settings. Among existing generalization methods, some works try to generalize to continuous outputs by capturing causal relations [32, 33]. Recent works like DDG [9] concentrate on capturing invariant semantic features, which might overlook the variational features for continuous predictions. In contrast, the meta-learning paradigm holds potential for regression settings due to its model-agnostic property and strong generalization ability.

The spearhead work MLDG [13] introduces MAML [12] into the domain generalization framework. [14] leverages class relationships and local sample clustering to capture the semantic features of different classes. These two operations are hard to be migrated to regression settings because the clustering assumption is usually not reasonable for regression. Moreover, in many regression tasks like age estimation, the semantic features might be unimportant, e.g., distinguishing each face might be useless for age regression. Instead, the style features, like the texture of the faces might be important information for age regression. Moreover, [30] proposes an implicit gradient to get stable meta-learning loss, which may provide orthogonal solution compared to our method.

## 3    Problem Setting and Notations

In this section, we introduce the formal definition of the DGR problem. We denote the input space and the label space by $\mathcal{X}$ and $\mathcal{Y}$, where $\mathcal{Y}$ has a continuous range from 0 to 1 and can include two sub-spaces, e.g., $\mathcal{Y}_{\text{source}}$ and $\mathcal{Y}_{\text{target}}$. $D_{\text{s}} = \{(\mathbf{x}, \mathbf{y}) \in \{\mathcal{X} \times \mathcal{Y}_{\text{source}}\}\}$ and $D_{\text{t}} =$

$\{(\mathbf{x}, \mathbf{y}) \in \{\mathcal{X} \times \mathcal{Y}_{\text{target}}\}\}$ respectively denote the source and target domain data. The model can only utilize $D_s$ at the training stage, and then predicts labels in $D_t$ without further adaptation. The above settings are very similar to the classification tasks of domain generalization. But the label spaces across domains are different in our regression setting. A prediction $\hat{y}$ from regression model $R$ can be denoted with $\hat{y} = R(x) = G(F(x))$. We use $F : \mathcal{X} \to \mathcal{Z}$ to denote a feature encoder, where $\mathcal{Z}$ is a feature space. After the encoder, we use a linear regressor with sigmoid activation to map the range of predictions into $[0, 1]$, i.e., $G : \mathcal{Z} \to \mathcal{Y}$.

## 4 Margin-Aware Meta Regression

### 4.1 Distribution Alignment Produces Regression Margin

Following the typical setting of domain generalization that domain labels are available. We split $D_s$ into $K$ source domains $\{D_1, D_2, \cdots, D_K\}$ and simulate the generalization setting between $D_s$ and $D_t$. As we know, feature alignment is the core idea of many typical domain alignment solutions for domain adaptation [34] as well as domain generalization [8]. For domain generalization, the alignment is usually performed among multiple source domains to find domain-invariant semantic features. This alignment can be formalized using a general discrepancy measure, i.e., *integral probability metric* (IPM, [35]). Let $X_1, X_2$ denote two independent random variables from domain distributions $\mathbb{P}_i$ and $\mathbb{P}_j$. The domain discrepancy can be defined with:

$$\text{IPM}(\mathbb{P}_i, \mathbb{P}_j) := \sup_{f \in \mathcal{H}} [\mathbb{E}[f(\mathbf{X}_1)] - \mathbb{E}[f(\mathbf{X}_2)]], \tag{1}$$

where $\mathbb{E}$ denotes the expectation, $f$ denotes the transformation function in function space $\mathcal{H}$. Applying specific condition on $\mathcal{H}$, IPM can be transformed into many popular measures, such as *maximum mean discrepancy* (MMD, [36]) and *wasserstein distance* (WD, [37]).

Incorporating the domain discrepancy between $\mathbb{P}_i$ and $\mathbb{P}_j$, the objective of the regressor can be formulated as:

$$\min_{\Theta} \sup_{\substack{(\mathbf{x}_1, \mathbf{y}_1) \in D_i, \\ (\mathbf{x}_2, \mathbf{y}_2) \in D_j}} \left[ L_{\Theta}(\mathbf{x}_1, \mathbf{y}_1) + L_{\Theta}(\mathbf{x}_2, \mathbf{y}_2) + \widehat{\text{IPM}}(\mathbf{x}_1, \mathbf{x}_2) \right], \tag{2}$$

where $\Theta$ is model parameter, $L_{\Theta}(\mathbf{x}, \mathbf{y}) = ||R_{\Theta}(\mathbf{x}) - \mathbf{y})||$ is the empirical risk and can be the squared loss, $\widehat{\text{IPM}}$ is the estimator from two batch examples $\mathbf{x}_1$ and $\mathbf{x}_2$. For example, $\widehat{\text{IPM}}$ can be the unbiased U-statistic estimator $\widehat{\text{MMD}}_u^2(\mathbf{x}_1, \mathbf{x}_2)$ [36]. In general domain generalization for classification tasks, all terms in the above objective could be minimized. However, our regression setting is like open domain generalization, which learns a model from the source domain and inferences in unseen target domains with novel classes [11]. To regress unseen target values, one strategy is to simulate the setting in the training stage. That means the labels in $D_i$ and $D_j$ have few or no overlaps. Therefore, when the domain discrepancy $\widehat{\text{IPM}}$ is minimized, there might be only one term minimized between $L_{\Theta}(\mathbf{x}_1, \mathbf{y}_1)$ and $L_{\Theta}(\mathbf{x}_2, \mathbf{y}_2)$. This problem can be formally introduced with the following definition:

**Proposition 1** (Regression Margin). *Let $(X_1, Y_1)$ and $(X_2, Y_2)$ be the random variables corresponding to two source domains $D_i, D_j$, the $[a, b]$ and $[c, d]$ be the regression interval of $Y_1, Y_2$. When $\widehat{\text{IPM}}$ is reduced to 0 for a function $f$, we have*

$$M_{i,j} = \inf |\mathbb{E}[f(\mathbf{X}_1) - Y_1] - \mathbb{E}[f(\mathbf{X}_2) - Y_2]| \tag{3}$$

$$= \inf |(\mathbb{E}[f(X_1)] - \mathbb{E}[f(X_2)]) + \mathbb{E}[Y_2 - Y_1]| \tag{4}$$

$$= \min(|c - b|, |a - d|). \tag{5}$$

The regression margin represents the minimal margin (or difference) between errors in the two domains (i.e., Eq. (3)). Eq. (4) is the rearrangement of Eq. (3). In Eq. (4), because $\widehat{\text{IPM}}$ is reduced to 0 for the function $f$, $\mathbb{E}[f(X_1)] - \mathbb{E}[f(X_2)] = 0$, then obtaining the Eq. (5). The above analysis suggests that a large domain margin $M_{i,j}$ can lead to a divergent optimization when simultaneously minimizing the domain discrepancy and the empirical risks. One strategy is to bypass explicit feature alignment. For example, in the meta-learning paradigm towards domain generalization, one can learn a meta-model by a bi-level optimization. In the inner optimization, the model learns on a support (source) domain. In the outer optimization, the learned model tries to generalize to a query (target)

domain. This training strategy naturally avoids explicit feature alignment. Moreover, the bi-level optimization emphasizes the importance of query loss, which might alleviate the above regression margin because the inner model and outer model can be viewed as different sampling instances in parameter space.

## 4.2 Regression Margin Leads to Sampling Bias in Meta-Learning

Existing meta-learning domain generalization methods are sub-optimal for the DGR problem. In the classification, each meta-task consisting of support tasks and query tasks is assumed to have the same sampling probability. However, the responding intervals of the support and query have ordinal relations in regression. When the regression margin between the support and query tasks is larger, the sampling probability is smaller. The left part of Fig. 2 depicts the relationship between the regression margin and the sampling strategies of meta-tasks. Intuitively thinking about the extreme case that when the regression margin is close to 1, the corresponding sampling probability of meta-tasks is close to 0. We formalize this using a simple theorem:

**Theorem 1** (Sampling Bias in Meta-Learning). *Given a support domain $i$, let $S_{(j|i)}$ denote the number of available query domain $j$ that can be sampled. Let $M_{i,j}^1, M_{i,j}^2$ denote the regression margin of the meta-task 1 and meta-task 2. if $M_{i,j}^1 > M_{i,j}^2$, then $S_{(j|i)}^1 < S_{(j|i)}^2$.*

The intuitive explanation is: the number of sampling strategies of a larger regression margin meta-task is always less than a small margin meta-task. We will provide a simple and intuitive proof below.

*Proof.* Following the previous description, the source data $D_s$ can be sorted into $K$ disjoint source domains $\{D_1, D_2, \cdots, D_K\}$ according to their regression interval. The query and support tasks are sampled from $D_i, D_j$ with regression interval $[a, b]$ and $[c, d]$ respectively. Let $\Delta$ denote the length of single regression interval, $n = \frac{M_{i,j}}{\Delta}$ denote the number of spanning intervals of regression margin $M_{i,j}$. Given a support task on domain index $i$, the query tasks on $j$-th domain have $S_{(j|i)}$ choices:

$$S_{(j|i)} = \begin{cases} K - (i + n), & \text{if } i \leq n \\ (i - n), & \text{if } i > K - n \\ K - 2n + 1, & \text{if } i > n \text{ and } i \leq K - n \end{cases} \tag{6}$$

From the above equation, when the regression margin $M_{i,j}$ increases (i.e., $n$ is increasing), the number of available-to-sample query tasks decreases, leading to a smaller number of eligible meta-tasks. $\square$

## 4.3 Margin-Aware Meta-Training

As illustrated by the left part of Fig. 2, a larger regression margin between the support and query tasks usually means a harder meta-task. Therefore, without any specialized sampling strategy, the meta model is prone to be *biased towards* the small margin tasks. To alleviate this issue, we want the large margin meta-task to have a larger weight in the meta-learning process. One direct strategy is to calculate the weight using the domain discrepancy, i.e., a larger regression margin means a larger meta-task weight. The learning objective can be redefined with:

$$\min_{\Theta} \sup_{\substack{(\mathbf{x}_q, \mathbf{y}_q) \in D_i, \\ (\mathbf{x}_s, \mathbf{y}_s) \in D_j}} L_{\Theta'}(\mathbf{x}_q, \mathbf{y}_q) \cdot d(\mathbf{x}_s, \mathbf{x}_q) \qquad s.t. \ \Theta' = \Theta - \beta \nabla_\Theta \left[ L_\Theta(\mathbf{x}_s, \mathbf{y}_s) \right], \tag{7}$$

where $D_i, D_j$ respectively denote the query domain and the support domain, $d$ is discrepancy functions like $\widehat{\mathrm{MMD}}_u^2(\cdot, \cdot)$ or simple Euclidean metric, and $\beta$ is the inner loop learning rate on the support domain $\{\mathbf{x}_s, \mathbf{y}_s\}$.

The graphical training process of one meta-task can be seen in the right part of Fig. 2. Different from existing meta-learning models, our MAMR model considers the domain discrepancy by discrepancy function $d(\cdot)$, but the data node in $d(\mathbf{x}_s, \mathbf{x}_q)$ does not have gradients. The reason is directly minimizing this domain discrepancy might harm the generalization ability of our MAMR model. Our task weighting method is similar to recent sharpness-aware minimization [38], which simultaneously minimizes loss value and loss sharpness. The related topic can also have an extension to penalizing gradient norm [39] and independence-driven importance weighting [40]. With Euclidean distance $d(\cdot)$, we describe the detailed method in Algorithm 1.

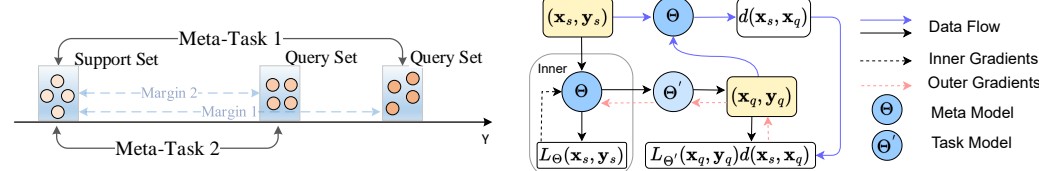

Figure 2: **Left:** The graphical illustration of the regression margin with sampling strategies of meta-tasks. **Right:** Our model's training process. Note that in the training process, meta-models share identical parameters $\Theta$, and the blue data flow does not involve gradient backpropagation.

---

**Algorithm 1** Training Algorithm of MAMR

---

**Input**: The source domains data $D_s$, the inner loop learning rate $\beta$, the out-loop learning rate $\alpha$, the domain number $K$ to split $D_s$, model parameters $\Theta$.
**Output**: The learned $\Theta$.

1: Split the source data $D_s$ into sub-domains $\{D_1, D_2, \cdots D_K\}$.
2: **while** not convergence **do**
3:     Sample $T = K(K-1)/2$ domain pairs $\{(D_i, D_j)\}$ that $i \neq j$.
4:     **for** $index = 0 \to T$ **do**
5:         Sample a batch of support data $(\mathbf{x}_s, \mathbf{y}_s) \in D_j$ and query data $(\mathbf{x}_q, \mathbf{y}_q) \in D_i$;
6:         Compute task discrepancies: $d(\mathbf{x}_s, \mathbf{x}_q) = ||F(\mathbf{x}_s) - F(\mathbf{x}_q)||_2$;
7:         Get task-specific model parameters: $\Theta' = \Theta - \beta \nabla_\Theta [L_\Theta(\mathbf{x}_s, \mathbf{y}_s)]$;
8:         Compute the weighted regression error: $L_{\Theta'}(\mathbf{x}_q, \mathbf{y}_q) \cdot d(\mathbf{x}_s, \mathbf{x}_q)$;
9:         Update $\Theta$: $\Theta = \Theta - \alpha \nabla_\Theta [L_{\Theta'}(\mathbf{x}_q, \mathbf{y}_q) \cdot d(\mathbf{x}_s, \mathbf{x}_q)]$;
10:     **end for**
11: **end while**

---

# 5 Experiments

In this section, we will empirically explore what MAMR can learn and compare it to related works from the view of performance and methodology, including introductions to baselines and experimental details, results on three datasets, and detailed analyses.

## 5.1 Baselines

We use multiple domain generalization and the variants of domain adaptation methods as baselines, including: (1) risk minimization methods (**ERM** [41], **IRM** [42]); (2) feature alignments and robust optimization (**MMD** [8], **CORAL** [43], **DANN** [34], **SD** [44], **Transfer** [45]), **MODE** [10]; (3) subspace alignments (**RSD** [3]); (4) self-supervised and data augmentation methods (**SelfReg** [46], **CAD** [47], **MTL** [48]) (5) meta-learning (**MLDG** [13]) and (6) disentanglement and causality method (**DDG** [9], **CausIRL** [49]). All the introductions of baselines can be seen in Appendix A.

## 5.2 Training and Evaluation

To ensure fairness and comparability, we put all the baselines into a public evaluation benchmark DomainBed [50]. For age regression, we uniformly use ResNet12 as the backbone encoder $F$ for all methods. ResNet12 is a popular encoder in meta-learning for few-shot learning. For rental regression, we uniformly use a 5-layer MLP as the backbone encoder $F$. For regressor $G$, we use a single linear neural network followed by a sigmoid function. Note that all labels are normalized from 0 to 1. Including toy experiments, all methods are implemented with Pytorch and can be executed on an NVIDIA RTX 3090 GPU. Appendix B provides detailed settings of the hyper-parameters, such as the learning rates, the training seeds, etc.

## 5.3 Toy Causality Dataset and Results

To figure out what the MAMR model can learn in regression problems, we create a toy dataset in which the input examples and their responding values obey some causal mechanism. We assume the

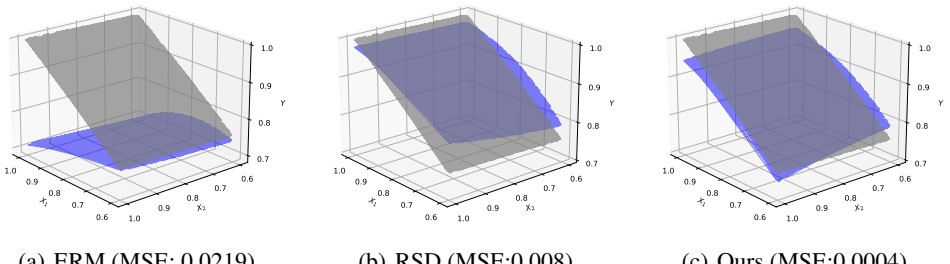

| (a) ERM (MSE: 0.0219) | (b) RSD (MSE:0.008) | (c) Ours (MSE:0.0004) |

Figure 3: The toy experiments illustrate the ground truth test landscape (gray color) and prediction regions (blue color). Each method's performance is reported with Mean Squared Error (MSE).

1-dimensional random variables $X_1$ and $X_2$ follow a uniform distribution in [0,1], and the responding values Y are under the control of $X_1$ and $X_2$. The control mechanism can be complex as given in Appendix C. At training stage, regression models can only use $X_1 \in [0, 0.6]$ and $X_2 \in [0, 0.6]$. At the test stage, we record the regression values when given $X_1 \in [0.6, 1]$ and $X_2 \in [0.6, 1]$.

The toy experiments sample 15000 and 10000 regression tasks at the training and test stage, respectively. We use a 4-layers fully connected neural network for ERM, RSD and our MAMR. Fig. 3 provides the test time explorations results of the three methods. On 10000 test tasks, the ground-truth responding values and the predicted values respectively form a gray region and a blue region. When given unseen values of $X_1$ and $X_2$, ERM fails to use the causal mechanism. The strong baseline method RSD captures a part of the causal mechanism. MAMR gets the best exploration performance by maximum causal discovery.

## 5.4 Cross-Domain Age Estimation Datasets

Perfect age estimation is based on the assumption that all age data are available, while many real-world datasets are not perfect and have partial ages due to privacy concerns. Hence age estimation has been introduced in cross-domain works [18, 51].

**CACD**[1]. Cross-Age Celebrity Dataset (CACD) contains 163,446 images from 2,000 celebrities collected from the Internet. The age of celebrities ranges from 16-62 and can be classified into 5 disjoint age intervals (domains), i.e., $[15-20), [20-30), [30-40), [40-50), [50-60]$. The images of each celebrity are sampled by different devices across multiple years. Therefore each domain has different facial characteristics. To consider the overlapped intervals, we further create **CACD-O** dataset, where each interval has 3 ages of neighbors, e.g., $[15-20)$ includes 8 different ages from 15 to 22 and $[20-30)$ has 15 ages from 18 to 32.

**AFAD**[2]. The Asian Face Age Dataset (AFAD) originally is an age estimation dataset containing more than 160K face images and aging labels. We split the dataset into 5 age intervals (domains), i.e., $[15-20), [20-25), [25-30), [30-35), [35-40]$. Like CACD, each age interval has its own face characteristics and can be viewed as 5 related domains for regression.

In each task, only one domain is viewed as the target domain, and the left is viewed as sources. Please refer to Appendix E for more details on these age estimation datasets.

## 5.5 Cross-Domain Rental Prediction Dataset

The Rental dataset [3] was released by an online competition in 2019 to predict housing rental in Shang Hai, China. The data categories include rental housing, regions, second-hand housing, supporting facilities, new houses, land, population, customers, real rent, etc. We split 15 regions into 4 groups as 4 different domains. Each domain has different rentals due to its population and economic conditions. Please refer to Appendix D for more introduction to this dataset.

---

[1]http://bcsiriuschen.github.io/CARC/

[2]https://afad-dataset.github.io/

[3]https://ai.futurelab.tv/contest_detail/3#contest_des

Table 1: Regression results on 4 cross-domain datasets with training-domain validation. The **"Average"** denotes the average Mean Squared Errors on 4 datasets. The "-" denotes not comparable results due to different architectures. The minimum values are **bolded**. Note that we set the standard variances to 0 if they are less than 0.001. **More performance details for each dataset can be seen in Appendix D and Appendix E.**

| Algorithms/Datasets | CACD | CACD-O | AFAD | Rental | Average |
|---|---|---|---|---|---|
| ERM ([41], 1998) | $0.0258_{\pm 0.001}$ | $0.0236_{\pm 0.000}$ | $0.0269_{\pm 0.000}$ | $0.0477_{\pm 0.003}$ | 0.0310 |
| IRM ([42], 2019) | $0.0368_{\pm 0.017}$ | $0.0256_{\pm 0.000}$ | $0.0285_{\pm 0.001}$ | $0.0496_{\pm 0.000}$ | 0.0351 |
| MLDG ([13], 2018) | $0.0260_{\pm 0.000}$ | $0.0235_{\pm 0.000}$ | $0.0268_{\pm 0.001}$ | $0.0465_{\pm 0.001}$ | 0.0307 |
| MMD ([8], 2018) | $0.0286_{\pm 0.000}$ | $0.0263_{\pm 0.000}$ | $0.0301_{\pm 0.000}$ | $0.0461_{\pm 0.000}$ | 0.0328 |
| CORAL ([43], 2016) | $0.0255_{\pm 0.000}$ | $0.0231_{\pm 0.000}$ | $0.0272_{\pm 0.003}$ | $0.0615_{\pm 0.019}$ | 0.0343 |
| DANN ([34], 2016) | $0.0269_{\pm 0.000}$ | $0.0259_{\pm 0.001}$ | $0.0290_{\pm 0.001}$ | $0.0474_{\pm 0.002}$ | 0.0323 |
| SD ([44], 2021) | $0.0248_{\pm 0.000}$ | $0.0227_{\pm 0.000}$ | $0.0270_{\pm 0.001}$ | $0.0493_{\pm 0.000}$ | 0.0598 |
| MTL ([48], 2021) | $0.1447_{\pm 0.000}$ | $0.1456_{\pm 0.000}$ | $0.2122_{\pm 0.001}$ | $0.0467_{\pm 0.001}$ | 0.1373 |
| SelfReg ([46], 2021) | $0.0252_{\pm 0.000}$ | $0.0232_{\pm 0.000}$ | $0.0281_{\pm 0.000}$ | $0.0526_{\pm 0.010}$ | 0.0323 |
| Transfer ([45], 2021) | $0.1446_{\pm 0.000}$ | $0.1379_{\pm 0.000}$ | $0.2122_{\pm 0.000}$ | $0.0475_{\pm 0.001}$ | 0.1355 |
| RSD ([3], 2021) | $0.0313_{\pm 0.000}$ | $0.0264_{\pm 0.000}$ | $0.0298_{\pm 0.000}$ | $0.0497_{\pm 0.005}$ | 0.0343 |
| CAD ([47], 2022) | $0.1447_{\pm 0.000}$ | $0.1849_{\pm 0.000}$ | $0.2122_{\pm 0.000}$ | $0.0555_{\pm 0.015}$ | 0.1493 |
| CausIRL ([49], 2022) | $0.0278_{\pm 0.000}$ | $0.0257_{\pm 0.002}$ | $0.0296_{\pm 0.000}$ | $0.0463_{\pm 0.000}$ | 0.0323 |
| DDG ([9], 2022) | $0.0490_{\pm 0.000}$ | $0.0268_{\pm 0.000}$ | $0.0302_{\pm 0.000}$ | — | — |
| MODE ([10], 2023) | $0.0283_{\pm 0.000}$ | $0.0268_{\pm 0.000}$ | $0.0299_{\pm 0.000}$ | $0.0464_{\pm 0.000}$ | 0.0329 |
| MAMR | $\mathbf{0.0189}_{\pm 0.000}$ | $\mathbf{0.0225}_{\pm 0.000}$ | $\mathbf{0.0238}_{\pm 0.000}$ | $\mathbf{0.0459}_{\pm 0.000}$ | **0.0278** |

## 5.6 Quantitative Comparisons

Comparison to **risk minimization** methods. ERM and IRM are typical risk minimization methods. From Tab. 1, we find that ERM is better than IRM, which might imply that the gradient invariance in IRM is useless for our problem. Another result is that the naive ERM is surprisingly comparable with advanced methods, e.g., MMD, DANN and MLDG. Even on AFAD dataset, ERM is a very strong baseline. Previous works [50, 52] also find a similar phenomenon in classification tasks.

Comparison to the methods using **feature alignments and robust optimization**. As discussed in Sec. 4, directly using feature alignments, e.g., MMD, DANN and CORAL, may perform poorly due to the regression margin. Furthermore, DANN and Transfer try to apply adversarial robustness, and MODE uses style augmentation for distribution robustness. Our results demonstrate the robustness design in these methods might bring the opposite impact on ordinal predictions.

Comparison to **subspace alignments**, e.g., RSD. We find that RSD gets comparable performance with respect to feature alignment methods. With principal angle alignment between sub-spaces, the sub-space alignments effectively slack the traditional feature alignments. This might imply that the domain adaptation method RSD can also generalize to out-of-distribution data.

Comparison to **self-supervised and data augmentation methods**, e.g., SelfReg. The self-supervised methods, especially with contrastive learning, can be strong baselines for our problem. The reason might be that SelfReg uses strong data augmentation and mixup operation in their models. We find the follow-up work CAD does not surpass SelfReg. The reason might be that the part of marginal distribution alignment in CAD harms the generalization ability like DANN. MTL augments the original feature space with the marginal distribution of feature vectors. However, MTL performs poorly in our regression settings. The reason might be augmenting the original feature space destroys the ordinal information of features.

Comparison to **meta-learning method**. MLDG simultaneously optimizes the support risks and query risks. While in DGR, the support and the query tasks usually change a lot, which makes the MLDG hard to be optimized. Our method does not simultaneously optimize the two risks and is attentive to hard tasks. The experiments also demonstrate that our method outperforms MLDG.

Comparison to **disentanglement/causality**. DDG disentangles the latent representations into semantic features and variation features. DDG may capture the causal mechanism between the inputs and their responding values. However, our further experiments with CausIRL method demonstrate that DDG can collapse with generated variational samples. DDG is originally proposed to minimize the semantic difference among generated samples from the same class while diversifying the variation across

Table 2: Ablation studies on CACD dataset with training-domain validation. Each regression interval (domain) denotes the target interval with the others as source intervals.

| Methods | [15-20) | [20-30) | [30-40) | [40-50) | [50-60] | Avg |
|---------|---------|---------|---------|---------|---------|-----|
| MAMR-   | $0.0348_{\pm 0.01}$ | $0.0284_{\pm 0.01}$ | $0.0015_{\pm 0.00}$ | $0.0156_{\pm 0.01}$ | $0.0235_{\pm 0.01}$ | 0.0208 |
| MAMR-G  | $0.0475_{\pm 0.00}$ | $0.0505_{\pm 0.03}$ | $0.0248_{\pm 0.02}$ | $0.0431_{\pm 0.02}$ | $0.0754_{\pm 0.04}$ | 0.0483 |
| MAMR-P  | $0.0331_{\pm 0.01}$ | $0.0143_{\pm 0.00}$ | $0.0021_{\pm 0.00}$ | $0.0078_{\pm 0.00}$ | $0.0371_{\pm 0.01}$ | 0.0189 |

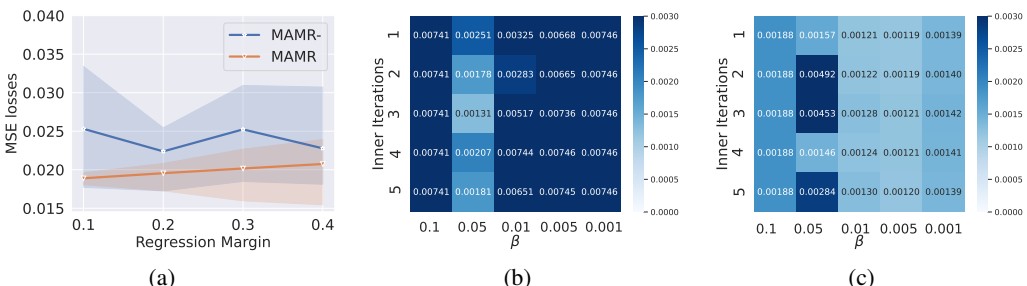

(a)          (b)          (c)

Figure 4: (a) The performances when changing regression margins. (b,c) The MSE heatmaps of regression tasks $[20, 30)$ and $[30, 40)$ in CACD by Oracle validation.

source domains. This design may let DDG overlook the variation features, which are coincidentally important in regression setting. Instead, CausIRL captures the style variables and finds sufficient conditions that do not rely on source domains.

## 5.7 Detailed Analyses

Tab. 2 provides 3 ablation models. MAMR- is our method without the margin-aware weighting mechanism. MAMR-G computes a mean weight for query tasks using the MMD with Gaussian kernel. MAMR-P computes the pair-wised Euclidean distances among the support and query tasks and provides a weight for each query task. We encourage MAMR-P to perform long range exploration by our proposed margin-aware weighting, which helps achieve better average regression performance. Besides that, the results demonstrate the averaged weight in MAMR-G may be invalid compared to pair-wised weights. The pair-wised Euclidean distances can be viewed as a special case of optimal transport distances [53] between the query data points and the support data points. Furthermore, Fig. 4(a) provides the regression performances of MAMR- and MAMR-P (MAMR). When manually enlarging the regression margin on the CACD dataset, MAMR consistently demonstrates better performance and smaller variance. Note that we set 0.1 as the start regression margin between the domain $[20, 30)$ and $[30, 40)$ in CACD.

The key hyper-parameters of the MAMR model include the inner loop learning rate $\beta$, the outer loop learning rate $\alpha$ and the iteration steps of the inner loop. To reduce the search of hyper-parameters, we set $\alpha = 0.1 * \beta$. We conduct a grid search for $\beta$ and the iteration steps. Fig. 4(b) and Fig. 4(c) provide the MSE heatmaps on the CACD dataset using two generalization tasks. We find that more inner iteration steps do not have a significant influence on the generalization results. This phenomenon is consistent with our analysis of the method: different from 5 or 10 inner steps in meta-learning for few-shot learning, fast adaptation by multi-steps is not necessary for DGR.

## 6 Conclusion and Limitations

We investigate domain generalization for ordinal regression problems. A margin-aware meta-learning regression method is proposed to achieve long-range exploration and interpolation. We build a regression benchmark to systematically investigate the performance of existing domain generalization methods for regression. Limitations: (1) Our empirical analyses demonstrate that domain generalization for regression still has a large exploration space when dealing with high-dimensional data. (2) Initial calculation of representation distance in meta-space is not reliable, one strategy is to consider a suitable warm-up strategy. (3) Finally, most used datasets have balanced source labels, applying MAMR to imbalanced source domains is also a more practical setting.

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
