# OpenReview forum: "Generalizing to Unseen Domains for Regression"
_NeurIPS.cc/2023/Conference — Submitted to NeurIPS 2023_

### Official Review · Reviewer_2irV · 2023-07-05

**Soundness:** 3 good
**Presentation:** 3 good
**Contribution:** 2 fair
**Rating:** 5
**Confidence:** 4

**Summary:**

The authors tackle domain generalization for regression tasks, or DGR. DGR is often different from classification tasks in DG as well as imbalanced regression tasks due to shift in labels.

Popular feature alignment type of algorithms using IPM may not be applicable to DGR because the labels of different domains may not align, which may cause a model to fit to only one domain.
The authors propose margin-aware meta learning framework to solve DGR by weighting query based on the distance to support, which mitigates sampling bias.

The authors provide experiment results on multiple datasets compared to several feature alignment methods in domain adaptation as well as self-supervised learning and meta-learning methods.

**Strengths:**

1. The authors clearly show the relationship between the margin between support and query, and hardness of query tasks.
2. From this analysis, the authors propose a simple weighting strategy for meta-learning in regression tasks.
3. Experiments are conducted on many datasets - toy, age estimation and cross-domain rental prediction.

**Weaknesses:**

1. The relationship between the margin and hardness that the authors demonstrated is not something new. It is somewhat obvious. For example, in Gaussian Processes, the uncertainty increases as a query point gets far away from support points.
2. Although the authors apply the proposed MAMR to meta learning framework, only one baseline model (MLDG) is used as a meta learning model. Without comparing it with multiple meta learning models e.g. ANIL, Neural Processes, it is hard to see if the improvement is coming from the proposed weighting or different framework (feature alignment vs. episodic meta training).
3. I believe MAMR is applied to MAML framework given that there is inner and outer loop updates (needs to be clarify unless it has already been specified). Related to 2, I would like to see if MAMR improves the performance on top of other meta learning algorithms. If my understanding is correct, MAMR should be applicable to any meta learning algorithms for regression tasks.
4. It could’ve been if different backbones were used for experiments.
5. In Line 324-327, the authors mentioned “different from 5 or 10 inner steps in meta-learning for few-shot learning, fast adaptation by multi-steps is not necessary for DGR.” It is not really true even for MAML. With some frameworks e.g. learn2learn, MAML performs as good with 1 step adaptation as 5 or 10 step adaptations.

**Questions:**

1. In line 168, what do you mean by divergent optimization?

**Limitations:**

Please refer to the weakness and questions above.

---

> ### Author Rebuttal · Authors · 2023-08-09
>
> ## Dear Reviewer 2irV:
> Thanks for your effort in giving detailed and constructive comments. Here are our point-to-point responses.
>
> > Weakness-1: *The relationship between the margin and hardness that the authors demonstrated is not something new.*
>
> **Response:** Thanks for your valuable comments. We agree that the relationship between the margin and hardness is straightforward, where a larger margin between support and query implies a more challenging task. Our innovation does not lie in emphasizing the discovery of this margin-hardness relationship, but highlighting the connection between the **margins** of regression labels and the **sampling probabilities** of hard tasks in the context of meta-learning. Specifically, we underline that these meta-tasks with larger label margins are prone to be overlooked by existing meta-learning algorithms, leading to a Sampling Bias (Theorem 1). This motivation has led us to design a margin-aware meta-learning algorithm.
> Our novelty also lies in practical regression settings from the view of DG: the domains with (i) overlapped or (ii) disjoint response intervals, fully covering scenarios where DG in regression is required.
>
> >  Weakness-2: * Although the authors apply the proposed MAMR to meta learning framework, only one baseline model (MLDG) is used as a meta learning model. Without comparing it with multiple meta-learning models e.g. ANIL, Neural Processes, it is hard to see if the improvement is coming from the proposed weighting or different framework (feature alignment vs. episodic meta training).*
>
> **Response:**
>
> **For more meta-learning methods.**
>
> 1.	Our work focuses on the DG setting, which usually needs **customized** meta-learning methods, e.g, MLDG. Unlike general meta-learning for few-shot learning, e.g., ANIL, in which it usually performs supervised adaptation at the test stage, under the DG setting, one model generally does not perform adaptation at the test stage. Therefore in the DG setting, the role of meta-learning is a robust learner to find optimal initialization for generalization, instead of rapid adaptation.
>
> 2. For not using (Conditional) Neural Processes (CNP). **Due to the absence of labels at the test stage of DG, it is hard to form the context of CNP.**  An approach is to treat the source data as the context, but this doesn't align with the intent of using context in CNP. CNP inherently assumes that the context is generally a part of the target data. The experiments in part C of the response PDF demonstrate that using source data as context does not have a significant advantage, even compared to the context with random noise. We are also trying to conduct experiments on more datasets, the results will be released once finished.
>
> 3. We adopt the well-known DG toolkit DomainBed, which selects MLDG as a strong meta-learning baseline designed for DG. Meanwhile, we observed that many meta-learning methods for DG do not release codes. **Even so, we tried to reproduce more meta-learning models in the framework of DomainBed, including ANIL, FOMAML, and their weighting versions. The results in part B of our response PDF demonstrate the strong performance of our method.**
>
> **For the improvement over baselines.**
> The improvement can be analyzed from two perspectives: episodic training and the weighting mechanism. Table 1 in our main paper demonstrates the superiority of using episodic training (MLDG and MAMR), compared with feature alignment (MMD, DANN). The ablation study of the weighting mechanism in Table 2 demonstrates its effectiveness. Furthermore, Figure 4a also shows the effectiveness of margin-aware weighting.
>
> >  Weakness-3: *I would like to see if MAMR improves the performance on top of other meta learning algorithms.*
>
> **Response:** Thanks for your constructive comments! We tested additional meta-learning algorithms, including ANIL and FOMAML. The results on the CACD dataset can be seen in the table of response PDF. The weighting versions demonstrate the effectiveness of our margin-aware weighting.
>
> >  Weakness-4: *It could’ve been if different backbones were used for experiments.*
>
> **Response:** Thanks for your suggestion. We employed three different encoders for three types of datasets, including a 4-layer MLP for the causality dataset, ResNet for age estimation, and a 5-layer MLP tailored for the rental dataset. We are trying additional encoder types such as ViT for the next version.
>
> >Weakness-5: *In Lines 324-327, the authors mentioned “different from 5 or 10 inner steps in meta-learning for few-shot learning, fast adaptation by multi-steps is not necessary for DGR.” It is not really true even for MAML. With some frameworks e.g. learn2learn, MAML performs as good with 1-step adaptation as 5 or 10 step adaptations.*
>
> **Response:** Under the  **few-shot setting**, we agree with that “MAML performs as good with 1-step adaptation as 5 or 10-step adaptations”. However, we are not a few-shot setting. At the training stage of our **DGR setting**, Figures 4b and 4c in our paper demonstrate more steps might hamper the optimal initialization of the model. Because more adaptation steps can alleviate the fitting difficulty for meta-model, which might lead to overfitting. Moreover, there is NO fast adaptation at the test stage in DGR, hence we said fast adaptation (by multi-steps) is not necessary for DGR.
>
> > Question-1: *In line 168, what do you mean by divergent optimization?*
>
> **Response:** The divergent optimization corresponds to convergent optimization. In our analysis of Proposition 1., when the features of two domains are aligned, the regression margin can still exist, resulting in non-zero training losses, which can lead to divergent optimization i.e., failed optimization for the model.
>
> **Thanks again for your effort in giving detailed and constructive comments. If you have more questions, feel free to discuss them with us. We are looking forward to your further messages in the discussion period.**

---

> ### Author Response · Authors · 2023-08-17
> **Additional experiments for Neural Process on CACD dataset**
>
> Dear Reviewer 2irV:
>
> In the discussion period, we accomplished additional experiment results for Neural Process on high-dimensional dataset CACD.  In this new experiment, we use the source data as the context of NP, to form its application in domain generalization setting.  To make a fair comparison, we add the used ResNet into the decoder and the encoder to reproduce NP.
>
> We found that the performance of NP is similar to MLDG, but much lower than our method.  As pointed out in our previous response,
> the reason is that NP methods inherently assume that the context is generally a part of the target data, hence it might not be a proper method for our DGR setting.
>
> | Methods | [15-20)      | [20-30)     | [30-40)      | [40-50)      |[50-60]      | Avg |
> |------------------|-----------------------|-----------------------|-----------------------|-----------------------|-----------------------|--------------|
> | **NP**          | ${0.0430}_{\pm 0.001}$ | ${0.0144}_{\pm 0.000}$ | ${0.0049}_{\pm 0.000}$ | ${0.0110}_{\pm 0.000}$ | ${0.0567}_{\pm 0.000}$ | $ 0.0260$  |
> | **MLDG**          | ${0.0454}_{\pm 0.000}$ | ${0.0140}_{\pm 0.000}$ | ${0.0028}_{\pm 0.000}$ | ${0.0137}_{\pm 0.000}$ | ${0.0540}_{\pm 0.000}$ | $ 0.0260$  |
> | **MAMR (ours)**   | ${0.0331}_{\pm 0.010}$ | ${0.0143}_{\pm 0.000}$ | ${0.0021}_{\pm 0.000}$ | ${0.0078}_{\pm 0.000}$ | ${0.0371}_{\pm 0.010}$ | $ {0.0189}$  |
>
> **How time flies. The discussion session is nearing its end. We are looking forward to your messages in this period. If you have more questions, feel free to discuss them with us.**
>
>   **It's possible you were previously occupied, or perhaps due to the lengthy review period, you've forgotten your initial comments. Regardless of the reason, thanks again for your participation to make NeurIPS a better conference.**
>
> Regards,
>
> Authors of Paper 4880

---

### Official Review · Reviewer_phkn · 2023-07-05

**Soundness:** 2 fair
**Presentation:** 2 fair
**Contribution:** 2 fair
**Rating:** 3
**Confidence:** 5

**Summary:**

This paper addressed the problem of domain generalization for regression, where the authors assume that different domains may have disjoint labeling space. The authors proposed a meta-learning-based method to learn the parameter initialization for DGR. In particular, the choosing frequency of different support and query domains relates to the label discrepancy. To evaluate the proposed method, the authors have also developed a DGR benchmark, considering both overlapping and nonoverlapping labels between the source and target domains.

**Strengths:**

1. Domain generalization in regression is a novel and interesting problem.

2. Leveraging task discrepancy to tackle domain generalization by meta-learning is novel.

3. The experimental results show effectiveness on a range of datasets.

**Weaknesses:**

1. It is unclear why using meta-learning to tackle the DGR problem in this paper. Although a range of works on DG has been proposed to use meta-learning, what are the significance and novelty of using meta-learning for domain generalization in regression?

2. The authors claim that the proposed method can enhance the exploration and interpolation capabilities. But it is unclear in the paper why the problem of domain generalization benefits from the interpolation capabilities.

3. The theoretical results in this paper are weak to me. Thm. 1 states that a larger regression margin of meta-tasks leads to more available domains. But it neglects the probability of meta-task over the set of labeling spaces.

4. the objective of meta-learning is a bi-level optimization problem.  It is unclear why not use the task discrepancy as one of the objectives of out-loop optimization, directly. Besides, how many iterations for the inner-loop optimization and outer-loop optimizations?

**Questions:**

Please answer my questions about the weakness during the rebuttal period. It is hopeful that my concerns can be addressed by the author's responses.

---

> ### Author Rebuttal · Authors · 2023-08-09
>
> ## Dear Reviewer phkn:
> Thanks for your effort in giving detailed and constructive comments. We hope your concerns can be successfully addressed by our point-to-point responses.
>
> > Weakness-1: *It is unclear why using meta-learning to tackle the DGR problem in this paper. Although a range of works on DG has been proposed to use meta-learning, what are the significance and novelty of using meta-learning for domain generalization in regression?*
>
> **Response:**  Thanks for your valuable comments. We will provide three paragraphs to introduce the novelty of our setting, the motivation for using meta-learning, and the differences between MAMR and other meta-learning methods for DG.
>
> 1. **Novelty of our setting.** Although several DG classification works claim that they can extend their methods to the regression scenarios, almost none of them realize that regression is a very different task compared to classification. Such significant difference mainly lies in that the ranges of the response intervals are often *different* due to distribution shift, even disjoint under extreme conditions. However, existing DG works implicitly assume that such ranges are the **same** in source and target domains, which is not reasonable in real-world applications. In our paper, we consider two practical regression settings: (i) the domains with overlapped response intervals; (ii) the domains with disjoint response intervals, fully covering scenarios where DG in regression is required.
>
> 2. **Motivation for using meta-learning.** It stems from the above regression setting (ii). In this scenario, the model is required to predict unseen regression values during testing. This is analogous to the existing meta-learning approaches designed for DG classification tasks, whose success motivates us to use meta-learning.
>
>
>  3. **Differences between MAMR and other meta-learning methods for DG.**  Our analysis and framework are both centered around the specific setting of regression. Compared to previous methods, we argue that all of them do not consider the ordering relationship between regression tasks. MLDG ([4], see the reference of global response) stands as the pioneer meta-learning approach in domain generalization for classification tasks.  MASF [5] represents another notable line of research on meta-learning for domain generalization. It leverages class relationships and local sample clustering to capture the semantic features of different classes. These two operations are hard to be migrated to regression settings because the clustering assumption is usually not reasonable for regression. Moreover, [6],  [7] and [8] tried to use learnable loss functions, feature critic modules, or model critic modules in the outer loop, but they similarly did not consider the ordinal relatedness. Lines 119-126 in main paper have discussed the relationship between our methods and existing meta-learning methods, we will add more discussions in the next version.
>
>
> >  Weakness-2: *For the explanation of interpolation in regression.*
>
> **Response:** Here we provide an example of interpolation task using age estimation. We split CACD into 5 domains according to the ranges of ages, i.e., (15, 20), (20-30), (30, 40), (40- 50), (50-60). If we select an in-between domain, e.g., (20-30), as the target domain, and the others as the source domains, then the target task can be called an interpolation regression task. Because the target domain labels are between the source domain labels.
>
> > Weakness-3: *The theoretical results in this paper are weak to me. Thm. 1 states that a larger regression margin of meta-tasks leads to more available domains. But it neglects the probability of meta-task over the set of labeling spaces.*
>
> **Response:**  Thanks for this comment.  We would like to clarify a potential misunderstanding for Thm.1: a larger regression margin of meta-tasks leads to **LESS** available domains. More importantly, Thm.1 is only intended to motivate our algorithm and we do not emphasize this theoretical contribution.
>
> 1. Most of the existing meta-learning methods for DG, e.g., MLDG, have an implicit assumption that the probability of meta-task over different domains follows a uniform distribution. We also assume a similar uniform distribution in our DGR setting.  If we know the distribution prior of the meta-task, we should be able to achieve better results.
>
> 2. Our experiments on the label imbalanced Rental dataset (see Figure 2 in Appendix) demonstrate that MAMR might handle some label-imbalanced scenarios. We will make them more clear in the next version.
>
>
> >  Weakness-4: *The objective of meta-learning is a bi-level optimization problem. It is unclear why not use the task discrepancy as one of the objectives of out-loop optimization, directly. Besides, how many iterations for the inner-loop optimization and outer-loop optimizations?*
>
> **Response:** Thanks for your detailed comments. The reason for not using task discrepancy as objective is that direct utilization will be equivalent to feature alignment. As pointed out in Proposition 1, feature alignment might not help to remove the regression margin, which is very different from the classification scenario. Our experiments using MMD or DANN in Table 1 also demonstrate feature alignment methods are suboptimal. Moreover, part A of the rebuttal PDF also provides a failure case when directly minimizing task discrepancy.
>
> For the steps of inner-loop optimization and outer-loop optimizations, we set 1 for the inner loop according to our hyper-parameter analysis in Sec 5.7. The steps of outer-loop optimizations are depended on the validation performance.  Figures 4b and 4c in the main paper illustrate detailed performances regarding the step of the inner-loop.
>
> **Thanks again for your effort in giving detailed and constructive comments. If you have more questions, feel free to discuss them with us. We are looking forward to your further messages in the discussion period.**

---

> ### Author Response · Authors · 2023-08-20
> **Feel free to start a discussion**
>
> Dear Reviewer phkn：
>
>    The discussion session is nearing its end. Have your concerns been addressed by our responses? If you have more questions, welcome to start a discussion with us.
>
>    Thanks again for your efforts and contribution.
>
>    Best regards,
>    Authors of Paper 4880

---

### Official Review · Reviewer_tR9s · 2023-07-07

**Soundness:** 3 good
**Presentation:** 3 good
**Contribution:** 3 good
**Rating:** 6
**Confidence:** 4

**Summary:**

This work investigates and analyzes domain generalization for regression tasks and propose a margin-aware meta-learning method for DG regression, which can help learn long-range exploration and interpolation. Also, a benchmark for evaluating regression dg is built.

**Strengths:**

- the paper is presented and derived well to make it easier to understand
- the difference between DG for regression and previous classification tasks is well analyzed


**Weaknesses:**

- is there any possibility to apply the proposed meta-learning based DG to more general regression tasks like object detection?

**Questions:**

Please refer to the weakness part

---

> ### Author Rebuttal · Authors · 2023-08-09
>
> ## Dear Reviewer tR9s:
> > Weakness-1: *Is there any possibility to apply the proposed meta-learning based DG to more general regression tasks like object detection?*
>
> **Response:** Thank you for your support of our work.　For more general regression tasks, e.g., object position regression and gaze direction regression, our method theoretically supports the above tasks. In fact, the RSD method [3] conducted the object position regression and gaze direction regression in its *domain adaptation* experiments. In these experiments, the prediction value is not a single value but a vector containing multiple elements like (x, y) in object position regression. The method performs feature-level alignment and uses the same MSE loss as single-value regression. In our domain generalization setting, the calculation of task discrepancy can be unrelated to the prediction dimensions, and the MSE loss can also be used to handle tasks involving multi-value regression.
>
> [3] "Representation Subspace Distance for Domain Adaptation Regression". ICML 2021.

---

### Official Review · Reviewer_JfrS · 2023-07-07

**Soundness:** 3 good
**Presentation:** 3 good
**Contribution:** 3 good
**Rating:** 7
**Confidence:** 4

**Summary:**

The paper describes an extension of domain generalization to the regression domain - traditionally, most of the research in Domain Generalization has occurred in image classification and to some degree in combined tasks such as object detection, depth estimation, and semantic segmentation and there has been a lack of research purely tackling domain generalization for regression. The authors specifically focus on the label shift aspect of domain shift, where the same input features might lead to two very different responses/regression values depending on the domain that they are in. The authors describe how the extra structure in regression problems (e.g. ordinal relatedness) is something that is mostly ignored by existing methods, and they propose a modification to the common meta learning methods for domain generalization to weight the meta-tasks appropriately based on this extra structure. The proposed method achieves SOTA among a variety of datasets in DomainBed.

**Strengths:**

### originality
* the paper is quite original. As the authors note, this problem has not been well studied, and so most of the results and discussion surrounding how to take advantage of domain generalization for regression is relatively novel.
* The technique proposed in the paper is fairly novel, even though it does not contain any individually novel pieces. Weighting a meta-learning strategy based off of the "difficulty" of the meta-task pair, defined by how related the two tasks are is certainly not something that I have seen before and is a neat strategy that could work in a variety of use-cases.
### quality
* The paper has mostly sound experiments, and does a good job providing both empirical and theoretical evidence to support its claims.
* The toy experiment (figure 3) is a great way to motivate why these kinds of techniques are needed and how existing methods may fail when they are presented with regression problems.
### clarity
* the paper is clear and easy to read, with both pseudo code, good diagrams/figures, and propositions that help clarify its points.
### significance
* domain generalization for regression is highly under-studied, and this work is a nice step forward to, at the very least, bring attention to this problem and how the underlying nature of it is quite different from classification.
* The technique of weighting the meta learning tasks based on some objective function feels like it could be used in a variety of different contexts, and I encourage the authors to explore this more in their conclusion/future work section.

**Weaknesses:**

The paper would benefit from a more nuanced discussion of the methods limitations, as well as more justification for some of the claims that are made. In particular,

* The limitations section is currently limited to 3 bullets, none of which have any justification - for example, "most used datasets have balanced source labels, applying to imbalanced is a more practical setting" - for this claim, what would be an example of an imbalanced dataset? Do we expect this kind of technique to still work there, or does the method fail on imbalanced source labels? Similar comments hold for the other limitations.

* Lines 36 - 38 talk about a potential outcome where feature alignment may be harmful for DGR tasks, but this would be much stronger if there was a toy experiment or example that the authors could talk about here that shows that this kind of feature alignment is a particular failure mode for popular feature alignment techniques.

* The paper specifically focuses on experiments that are mostly single value regression tasks in nature, e.g. rental prediction - however, a large portion of regression tasks have image features as input - for example, depth estimation or bounding box regression. The paper would benefit from a discussion on whether or not these techniques could extend to those more common regression tasks, or what would need to change in order to make that happen.

**Questions:**

* What is the compute required to run the meta learning algorithms? Does computing the margins cause significant extra weight?

* I know the limitations touch on "unreliable calculations of representation distances in meta space" - how sensitive is this technique to ensuring that these calculations are correct?

**Limitations:**

Addressed in the weaknesses section.

---

> ### Author Rebuttal · Authors · 2023-08-09
>
> ## Dear Reviewer  JfrS:
> We appreciate your support of our work and your efforts in giving detailed and constructive comments. We hope your concerns can be successfully addressed by our point-to-point responses.
>
> > Weakness-1: *The limitations section is currently limited to 3 bullets, none of which have any justification - for example, "most used datasets have balanced source labels, applying to imbalanced is a more practical setting" - for this claim, what would be an example of an imbalanced dataset? Do we expect this kind of technique to still work there, or does the method fail on imbalanced source labels? Similar comments hold for the other limitations.*
> >
> **Response:** Thanks for this detailed comment.  For justification of **imbalanced dataset**, we indeed have considered a type of label imbalance scenario in Rental dataset. In this dataset, most rental prices locate in a small range of price intervals (see Figure 2 in Appendix). The experiments show the competitive performance of our method. However, the imbalance pattern of target domain is similar to souce domains in this dataset. It is also very challenging when the imbalance patterns are different between the source and the target.
>
> For justification of **regression on high-dimensional data**. A direct empirical observation is that the regression error of high-dimensional data (age estimation dataset) is larger than low-dimensional data (causality dataset). Note that all target values have been cast into [0,1] for the two datasets.
>
> For justification of **initial calculation of representation distance**. In implementation, we perform a simple warm-up training for MAMR, that is, we do not use the weighing mechanism before some iterations, then the weighting mechanism can be introduced in the subsequent iterations. A better warm-up strategy can also be introduced in the future. We will add more related discussion in next version.
>
>
> > Weakness-2: *Lines 36 - 38 talk about a potential outcome where feature alignment may be harmful to DGR tasks, but this would be much stronger if there was a toy experiment or example that the authors could talk about here that shows that this kind of feature alignment is a particular failure mode for popular feature alignment techniques.*
> >
> **Response:** We appreciate the reviewer’s constructive comments. To make stronger evidence, we use MMD as an auxiliary loss for feature alignment on the toy dataset of our paper. The first three figures in our response PDF demonstrate the weights (0.0, 0.1, 1) of MMD loss corresponding to the prediction loss at the training stage. We can find that the MMD loss makes the test loss increase after some iterations, which can be viewed as the failure mode of MMD alignment.
>
> > Weakness-3: *The paper specifically focuses on experiments that are mostly single value regression tasks in nature, e.g. rental prediction - however, a large portion of regression tasks have image features as input - for example, depth estimation or bounding box regression. The paper would benefit from a discussion on whether or not these techniques could extend to those more common regression tasks, or what would need to change in order to make that happen.*
>
> **Response:** Thanks for this constructive comments. For image features as input, we have used age estimation tasks, in which the input is face images. Because our model calculates the task discrepancy in the feature level, it can be theoretically applied to the multi-value regression tasks like RSD method. Due to time limitation for experiments, we put it as a future work for multi-value regression tasks.
>
>
> > Question-1: *What is the compute required to run the meta learning algorithms? Does computing the margins cause significant extra weight?*
>
> **Response:** The computation overload of MAMR is similar to MAML from the view of implementation and optimization. On the one side, compared with standard MAML, we use fewer inner steps and create shorter computation graphs.  On the other side, although we use extra weight, the computation of extra weight does not generate a computation graph in Pytorch (the gradient backpropagation is usually time-consuming in deep learning framework), hence does not involve gradient backpropagation ( see the right part of Figure 2 in our main paper ).
>
> > Question-2: *I know the limitations touch on "unreliable calculations of representation distances in meta space" - how sensitive is this technique to ensuring that these calculations are correct?*
>
> **Response:** Thanks for this question. The sensitivity can be influenced by two direct factors: the warm-up training and the model initialization. To increase the quality of calculation, we perform a simple warm-up training for MAMR, that is, we do not use the weighing mechanism before some iterations, then the weighting mechanism can be introduced in the subsequent iterations. A better warm-up strategy can also be introduced in the future.  It's very obvious that different ways of initializing model parameters may result in varying initial representation distances. For Resnet, we suggest the default method in our version of Domainbed for regression, using normal distribution initialization for parameters.

---

> > ### Comment · Reviewer_JfrS · 2023-08-15
> > **Response**
> >
> > Thank you for responding to my concerns - I think the experiments and justifications make sense, and appreciate the time to write these comments. As long as the discussion from this response is in the camera ready version of the paper (e.g. the extended discussion on limitations, extra experiments, etc.), I am happy to move my score up to a 7 here.

---

> > > ### Author Response · Authors · 2023-08-16
> > > **Thanks for raising your score to 7!**
> > >
> > > Dear Reviewer JfrS,
> > >
> > > Many thanks for your support and raising your score to 7. We will include all of our discussions in our paper.
> > >
> > > Best regards,
> > >
> > > Authors of Paper 4880

---

### Official Review · Reviewer_ajvr · 2023-07-07

**Soundness:** 4 excellent
**Presentation:** 3 good
**Contribution:** 4 excellent
**Rating:** 7
**Confidence:** 4

**Summary:**

This paper focused on the generalization problem in the regression context and proposed an interesting setting, Domain Generalization in Regression (DGR). Compared to domain generalization for classification, DGR is a more challenging task due to its different domain shift forms between source and target domains. Compared with the traditional domain generalization for classification where the source and target domains usually shared the same label set, the range of response variables between the source and target domains in the context of regression can be very different. From this perspective, this setting proposed in this paper is novel and worthy to be explored. This paper proposed to solve the DGR problem through the meta-learning philosophy. However, the authors pointed out that when applying the meta-learning method for the DGR problem, the harder meta-tasks with larger regression margins on the label discrepancy can be less sampled and this sampling bias made these harder tasks difficult to optimize. Then, the authors proposed to address this sampling issue by measuring the feature discrepancy between the query and support samples and assigning higher weights to harder meta-tasks. Extensive experiments showed that the proposed MAMR can solve the DGR problem in an effective manner.

**Strengths:**

1. This paper proposed a novel and practical setting, Domain Generalization in Regression. The generalization problem in the regression context was less explored in the community. This paper may inspire more explorations into this topic.
2. This paper analyzed the differences between the domain generalization tasks in the classification and the regression and showed the specific features of this problem, e.g., the distribution shift between the source and target domains can be under a different form, the sampling bias due to the ordinal relatedness in the regression. Based on these analyses, we can see that some existing methods cannot be directly applied to this field.
3. The motivation for the proposed methods is clear. Due to the sampling bias, the author proposed a weighted meta-learning method. The solution is reasonable.
4. The extensive experiments on some synthetic and real-world datasets demonstrated the effectiveness of the proposed MAMR.
5. This paper is well-written and easy to follow.

**Weaknesses:**

1. Some assumptions in the theoretical analysis can be further discussed. For example, it seems that Theorem needs to assume that the data from a meta-task distribute uniformly with respect to the response variable $y$.

**Questions:**

- In order to make Theorem 1 hold, do we need to assume the label distribution within a meta-task need to be uniform? If so, is it possible to extend the conclusion into imbalanced scenarios?
- In Section 5.3, it is interesting to see that MAMR performed better for the causality discovery in this toy example. Could the author explain more about the reasons why MAMR had better performances under this kind of scenario?
- In Table 2, it seems that the MAMR-G using MMD with Gaussian kernel usually had poorer performances compared to MAMR- (without any margin-aware weight strange). Could the author explain more about this interesting observation?

**Limitations:**

No severe limitations.

---

> ### Author Rebuttal · Authors · 2023-08-09
>
> ## Dear Reviewer ajvr:
> We appreciate your support of our work and your efforts in giving detailed and constructive comments. We hope your concerns can be successfully addressed by our point-to-point responses.
>
> >Weakness-1: *Some assumptions in the theoretical analysis can be further discussed. For example, it seems that Theorem needs to assume that the data from a meta-task distribute uniformly with respect to the response variable.*
>
> **Response:** Most existing meta-learning methods for DG，e.g., MLDG, have an implicit assumption that the probability of meta-task over different domains follows a uniform distribution. They did not know the distribution prior for the target domain, so as a compromise solution, they assume that the data distributions as well as the label distributions between the source and the target are similar. We also assume a similar uniform distribution in our DGR setting. The 3-th row of Algorithm 1 demonstrates our implicit assumption when sampling meta-tasks, and we will make it more clear in the next version. Even so, our experiments on the label imbalanced rental dataset (see Figure 2 in Appendix) demonstrate that our method might handle some imbalanced distributed scenarios.
>
> >Question-1: *In order to make Theorem 1 hold, do we need to assume the label distribution within a meta-task need to be uniform? If so, is it possible to extend the conclusion into imbalanced scenarios?*
>
> **Response:** Thanks for your constructive questions. Yes, we do. Making label distribution within a meta-task uniform is an implicit assumption of most meta-learning works for domain generalization like MLDG. Theoretically extending the conclusion into imbalanced scenarios is a very interesting and challenging work. For the imbalanced scenarios, we empirically evaluated our method on the label imbalanced Rental dataset (please refer to Figure 2 in the Appendix). We set the theoretical analysis for imbalanced scenarios as one of our future works.
>
> > Question-2: *In Section 5.3, it is interesting to see that MAMR performed better for the causality discovery in this toy example. Could the author explain more about the reasons why MAMR had better performances under this kind of scenario?*
>
> **Response:** Thanks for this insightful question. In Section 5.3, we adopt an assumption that better causal discovery benefits regression exploration.  Hence the toy dataset is designed with input X and their causal responding output Y. In our experiment, we designed a set of arithmetic reasoning mechanisms to generate Y from X （see Figure 1 in Appendix）.  The exploration performances of three types of methods, i.e., ERM, feature alignment by RSD, and our MAMR, demonstrate that MAMR can fit the arithmetic reasoning mechanisms more precisely and might go beyond pure correlation-based inference. Some research works specifically employ meta-learning for causal discovery, e.g., [1] and [2]. The RSD method can be grouped into the feature alignment category, whose drawback for DGR has been theoretically discussed in Proposition 1.
>
>
> > Question-3: *In Table 2, it seems that the MAMR-G using MMD with Gaussian kernel usually had poorer performances compared to MAMR- (without any margin-aware weight strange). Could the author explain more about this interesting observation?*
>
> **Response:** MAMR-G with Gaussian kernel has a key hyper-parameter, i.e., the variance $\sigma$, which is usually sensitive to tensor values. Choosing proper $\sigma$ is a laborious work, hence we adopt the default and standard usage of DomainBed that sets multiple $\sigma$ ( [0.001, 0.01, 0.1, 1, 10, 100, 1000]) and calculate the sum of the distance corresponding to these values. From the performance results, we observe that the standard setting of $\sigma$ lead to degenerated performance compared with MAMR-.
> As we know, MAML itself is difficult to train, and complex distance calculation methods like using Gaussian kernel may not necessarily benefit stable training. In contrast, without the hyper-parameter, the Euclidean distance stabilizes the optimization, getting better performance.

---

> > ### Comment · Reviewer_ajvr · 2023-08-18
> > **Response to Author Rebuttal**
> >
> > Thanks for the effort from the authors in answering my questions. My concerns have been addressed. And I will keep my current score and support the acceptance of this work.

---

> > > ### Author Response · Authors · 2023-08-18
> > > **Thanks for the score of 7 for "Generalizing to Unseen Domains for Regression"！**
> > >
> > > Dear Reviewer ajvr:
> > >
> > > We are happy that your concerns have been addressed! We will add the related discussion to our next version.
> > >
> > > Our paper still gets diverged ratings now, but the author-reviewer discussion period is nearing its end. Hence we are in need of feedback from the left three reviewers and welcome further discussion. We are also looking forward to your support in the following period.
> > >
> > > Thanks again for your contribution to bringing a better NeurIPS!
> > >
> > > Regards,
> > >
> > > Authors of Paper 4880

---

### Author Rebuttal · Authors · 2023-08-09

## Global Response:
Thanks for the efforts of reviewers, we are encouraged by the detailed and constructive comments.

We have provided comprehensive responses to the comments from each reviewer. Additionally, we have prepared a **one-page PDF** containing the necessary justification experiments for our claim. We hope that our responses address the reviewers' concerns. **If you have any further questions, please feel free to discuss them with us within the system.**

Here we list the necessary information used in the response for each reviewer, although some of them have been listed in our paper.

[1] "Meta Learning for Causal Direction". AAAI 2021.

[2] "A Meta-Reinforcement Learning Algorithm for Causal Discovery". CLEAR 2023.

[3] "Representation Subspace Distance for Domain Adaptation Regression". ICML 2021.

[4] "Learning to generalize: Meta-learning for domain generalization". AAAI 2018.

[5] "Domain Generalization via Model-Agnostic Learning of Semantic Features". NeurIPS 2019.

[6] "Loss Function Learning for Domain Generalization by Implicit Gradient". ICML 2022.

[7] "Feature-Critic Networks for Heterogeneous Domain Generalisation". ICML 2019.

[8] "Metareg: Towards domain generalization using meta-regularization". NeurIPS 2018.

[9] "On first-order meta-learning algorithms".  Arxiv 2018.

[10] "Rapid learning or feature reuse? towards understanding the effectiveness of maml". ICLR 2020.

---

### Decision · Program_Chairs · 2023-09-21

**Decision:**

Reject

**Comment:**

This paper studies the problem of Domain Generalization for regression and proposes a novel weighted meta-learning method. Some theoretical properties on the margin are presented. An experimental evaluation is provided with different datasets and baselines for comparisons.

In the reviews, the setting considered domain generalization for regression was evaluated as very interesting and promising.
Some reviewers ver like the contribution (Reviewers ajvr and jfrS).
Reviewer tR9s was rather positive in the initial review.
Reviewers phkn and 2irV were rather negative in their initial reviews.

During the rebuttal, the authors have provided some answers to the remarks raised by authors. They in particular corrected a typo related to Thm1 and indicate that "larger regression margin of meta-tasks leads to LESS available domains", as well as other precision in specific answers to reviewers. In the additional pdf, they also added new experimental results.

After rebuttal, there was a still a divergence between reviewers point of views.
Overall, I find that the theoretical results, while interesting, are a bit weak and in a sense rather expected. I do not see a strong result here in the context of Domain Generalization regression.
While the paper is borderline, I evaluate the contribution as being limited in its current form and I propose rejection.

However, I encourage the authors to work on the weaknesses raised to improve the submission for another venue.